# Direct Single-Operator Cholangioscopy and Intraductal Ultrasonography in Patients with Indeterminate Biliary Strictures: A Single Center Experience

**DOI:** 10.3390/diagnostics14131316

**Published:** 2024-06-21

**Authors:** Marco Sacco, Marcantonio Gesualdo, Maria Teresa Staiano, Eleonora Dall’Amico, Stefania Caronna, Simone Dibitetto, Chiara Canalis, Alessandro Caneglias, Federica Mediati, Rosa Claudia Stasio, Silvia Gaia, Giorgio Maria Saracco, Mauro Bruno, Claudio Giovanni De Angelis

**Affiliations:** 1Endoscopy Unit, Gastroenterology Department, AOU Città della Salute e della Scienza di Torino, 10126 Turin, Italy; mariateresastaiano@gmail.com (M.T.S.); eledallam@gmail.com (E.D.); stefaniacaronna@gmail.com (S.C.); simo.dibitetto@gmail.com (S.D.); canalis.chiara@gmail.com (C.C.); alessandrocaneglias13@gmail.com (A.C.); mediatifederica@gmail.com (F.M.); stasio.rosa.claudia@gmail.com (R.C.S.); silvia.gaia74@gmail.com (S.G.); giorgiomaria.saracco@unito.it (G.M.S.); mabru1964@gmail.com (M.B.); eusdeang@hotmail.com (C.G.D.A.); 2Endoscopy Unit, Gastroenterology Department, Section of Gastroenterology II, National Institute of Research IRCCS “Saverio De Bellis”, 70013 Castellana Grotte, Italy; marcantoniogesualdo@gmail.com

**Keywords:** cholangioscopy, indeterminate biliary stricture, ERCP, IDUS, biliary stenosis, cholangiocarcinoma

## Abstract

The evaluation of biliary strictures poses a challenge due to the low sensitivity of standard diagnostic approaches, but the advent of direct single-operator cholangioscopy (DSOC) has revolutionized this paradigm. Our study aimed to assess the diagnostic performance of DSOC and DSOC-targeted biopsies, intraductal ultrasound (IDUS), and standard brush cytology in patients with indeterminate biliary strictures (IBS). We reviewed patients who underwent advanced diagnostic evaluation for IBS at our endoscopy unit from January 2018 to December 2022, all of whom had previously undergone at least one endoscopic attempt to characterize the biliary stricture. Final diagnoses were established based on surgical pathology and/or clinical and radiological follow-up spanning at least 12 months. A total of 57 patients, with a mean age of 67.2 ± 10.0 years, were included, with a mean follow-up of 18.2 ± 18.1 months. The majority of IBS were located in the distal common bile duct (45.6%), with malignancy confirmed in 35 patients (61.4%). DSOC and IDUS demonstrated significantly higher accuracies (89.5% and 82.7%, respectively) compared to standard cytology (61.5%, *p* < 0.05). Both DSOC visualization and IDUS exhibited optimal diagnostic yields in differentiating IBS with an acceptable safety profile.

## 1. Introduction

Indeterminate biliary strictures (IBS) are regarded as such when the standard diagnostic workup turns out to be inconclusive, representing a diagnostic challenge for physicians [1]. Standard diagnostic workup includes cross-sectional imaging with computer tomography (CT)-scan and magnetic resonance imaging (MRI), endoscopic retrograde cholangiopancreatography (ERCP) with brush cytology and/or endoscopic ultrasound (EUS) with fine needle aspiration (FNA) or biopsy (FNB).

Despite using a combination of these techniques, there is a substantial risk of misdiagnosis: biliary strictures remain indeterminate in up to 20% of cases, and one in four surgically resected IBS cases exhibit benign histology [1,2].

The limitations of standard diagnostic workup can lead to multiple, repeated procedures to establish a diagnosis, leading to delays in treatment initiation (which may reduce the likelihood of curative resection in malignancy cases) and increased risks for procedure-related adverse events (AEs) [3,4,5,6].

In recent decades, several new diagnostic techniques have been developed to overcome these challenges. Single-operator cholangioscopy (SOC) and the evolution to digital cholangioscopy (DSOC) enable direct visualization of the biliary mucosa and the evaluation of cholangioscopic features of malignancy, with the added capability of obtaining tissue samples under direct visualization [7].

Intraductal ultrasound (IDUS) can obtain real-time, cross-sectional images of ductal and periductal structures using a high-frequency ultrasound probe inserted directly into the bile duct over a guidewire. IDUS findings consistent with malignancy include asymmetric wall thickening, disruption of layers, enlarged lymph nodes, and hypoechoic sessile masses or nodules [8], while benign strictures typically exhibit regular layering, smooth margins and homogenous and symmetric hyperechoic thickening of the biliary wall [9]. Despite promising data from the literature, the utilization of IDUS remains limited to a few centers.

Given the low sensitivity of standard diagnostics, some referral centers adopt a “one shot” approach, combining ERCP and multiple ancillary techniques (IDUS, cholangioscopy and cholangioscopy-based tissue acquisition, confocal laser endomicroscopy) in a single-endoscopic procedure, to maximize diagnostic yield and minimize the need for multiple interventions [10].

The aim of this study was to evaluate the diagnostic performance (sensitivity, specificity, accuracy, positive and negative predictive values) of DSOC, DSOC-guided biopsy, IDUS, and standard brush cytology in patients with IBS.

Secondary aims included assessing the safety profile of the procedure and evaluating the impact of previous stenting on diagnostic accuracy.

## 2. Materials and Methods

This is a single-center, retrospective, observational study on consecutive patients with IBS who underwent cholangioscopy and IDUS at the Endoscopy Unit of AOU Città della Salute e della Scienza di Torino (Turin, Italy) from January 2018 to December 2022.

We included patients who had previously undergone inconclusive diagnostic workups for biliary stricture, including ERCP with brushing cytology and/or EUS ± FNA/FNB, either performed at other hospitals or at our Unit.

Data on demographics, clinical history, endoscopic findings, histology, and follow-up were extracted from a prospectively collected database.

Inclusion criteria included the following: (1) patients with IBS undergoing DSOC for IBS between January 2018 and December 2022; (2) patients ≥ 18 years of age.

Exclusion criteria were as follows: (1) indication of DSOC other than IBS; (2) coagulation disorder with a contraindication to invasive endoscopic maneuvers (INR > 1.6, platelet count < 40 × 10^3^/mm^3^); (3) refusal to provide informed consent for the procedure and/or to the study.

All procedures were performed by highly experienced biliopancreatic endoscopists proficient in both EUS and ERCP. Prior to procedures, clinical information relevant to patient history was disclosed to the endoscopists. ERCP was performed in standard fashion with a TJF-180 duodenoscope (Olympus, Tokyo, Japan) with Propofol-induced deep sedation in the left lateral decubitus position. Antibiotic prophylaxis (beta-lactams or fluoroquinolones) was administered before cholangioscopy, and rectal indomethacin or diclofenac (100 mg) was routinely given for post-ERCP pancreatitis prophylaxis unless contraindicated. Patients were monitored in the hospital for at least one night, and blood tests, including lipases, were performed 6 and 24 h post-procedure.

IDUS examination was carried out with the introduction of a 20 MHz wire-guided miniprobe (UM-DP20-25R, Olympus, Tokyo, Japan) above the stricture and then gently passed through the stricture for ultrasonographic assessment.

Cholangioscope for DSOC (SpyGlass Discover™, Boston Scientific, Natick, MA, USA) was introduced over a guidewire up to the stricture for visual inspection.

DSOC-guided biopsies were performed with dedicated forceps (SpyBite™, Boston Scientific, Natick, MA, USA) to target suspicious areas of the IBS, ensuring adequate sampling with at least four visible tissue samples collected. Specimens were placed in plastic containers and fixed in 10% formaldehyde solution.

Brush cytology was performed under fluoroscopic view using a standard brush (Cytomax II Double-lumen Cytology Brush, Cook Medical, Bloomington, IN, USA).

### 2.1. Definitions

For DSOC-based diagnosis, the presence of either nodular masses or papillary projections, tortuous dilated vessels, and fragile, irregular mucosa were considered typical features of neoplastic strictures. Conversely, lesions were considered benign if they exhibited a flat surface, a fine network of vessels, a regular granular appearance, and/or smooth, non-fragile mucosa [11,12]. The final impression diagnosis of a benign or malignant stricture was determined during the procedure (Figure 1).

The IDUS-based diagnosis was made using previously published criteria: findings such as asymmetric wall thickening, disruption of layers, enlarged lymph nodes, and/or hypoechoic sessile masses or nodules were indicative of a malignant stricture [8].

Malignancy was confirmed by surgical specimen (when available), biopsy, or cytology demonstrating malignant cells. A benign diagnosis required a minimum of 12 months of clinical and radiological follow-up with no evidence of masses or progression on imaging, repeated sampling, or death.

The length of follow-up was calculated as the time between the procedure and surgery in patients undergoing surgical resection. For others, it was measured from the procedure date to death or the last clinical contact.

AEs were investigated using electronic health records and categorized based on onset (preprocedural, intraprocedural, post-procedural within <14 days and late as >14 days) and severity (mild, moderate, severe, and fatal) [13].

### 2.2. Statistical Analysis

Demographic, clinical, procedural, and pathology details were summarized as the mean and standard deviation (±SD) for continuous variables and count and percentage for categorical variables.

Operating characteristics, including sensitivity, specificity, positive predictive value, negative predictive value, and diagnostic accuracy, were calculated for each diagnostic technique.

Fisher’s exact test was utilized to compare the operating characteristics, and the Student’s *t*-test was used to compare continuous variables across subgroups. A *p*-value of 0.05 was considered to be statistically significant.

Statistical analysis was performed with MedCalc Statistical Software version 19.2.6 (MedCalc Software bv, Ostend, Belgium).

The study protocol and consent form were approved by the local Institutional Review Board, and the study was conducted according to the Declaration of Helsinki.

## 3. Results

A total of 57 patients were included, mostly male (39 patients, 68.4%) with a mean age of 67.2 ± 10.0 years. The majority of patients had comorbidities (34 patients, 59.6%), particularly cardiovascular diseases (24 patients, 42.1%), chronic liver diseases (7 patients, 12.3%), and pulmonary diseases (6 patients, 10.5%).

Only one patient had a previous diagnosis of primary sclerosing cholangitis, and 11 patients (19.3%) had a history of previous or active tobacco consumption.

All patients had at least one previous inconclusive diagnostic procedure: 39 patients had previous ERCPs (68.4%), with a mean number of previous procedures of 2.0 ± 1.4; 14 patients had a previous attempt of IBS characterization with EUS ± FNA/FNB (24.6%) and 4 patients (7.0%) had a previous attempt with percutaneous ultrasound transhepatic biliary tissue acquisition; 38 patients (66.7%) had a previous biliary sphincterotomy and 30 patients (52.6%) had a biliary stent in place.

Strictures were distributed throughout the biliary tree; the most common location of IBS was the distal common bile duct (26 patients, 45.6%), followed by the common hepatic duct (13 patients, 22.8%), hepatic hilum (12 patients, 21.1%), intrahepatic ducts (5 patients, 8.8%), and the cystic duct (1 patient, 1.7%). The baseline characteristics of patients are presented in Table 1.

DSOC was successfully performed in all patients, with 52 patients undergoing DSOC-guided tissue acquisition. DSOC-guided biopsies were not performed in five patients: two due to the evidence of extra-ductal lesions and three due to technical failure in retrieving the forceps from the cholangioscope channel. In these three patients, brush cytology was the only method of tissue acquisition used. Therefore, the technical success rate of DSOC-guided biopsy was 94.5%.

Fifty-two patients (91.2%) underwent IDUS. Passage of the miniprobe through the stricture was not possible in three patients, while in the other two cases, IDUS was temporarily unavailable, resulting in a technical success rate of 94.5%. Additionally, 39 patients underwent brush cytology; it was not performed in 3 patients due to the evidence of extra-ductal lesions and in 15 patients because specimens collected by DSOC-guided biopsies were deemed sufficient.

The final diagnosis was consistent with malignancy in 35 patients (61.4%), with cholangiocarcinoma being the most common etiology (77.1%), followed by intraductal papillary biliary neoplasm with high-grade dysplasia (4 patients, 11.4%), ampullary adenocarcinoma (3 patients, 8.6%), and pancreatic cancer (1 patient, 2.9%). Malignancy was confirmed by surgical pathology in 22 patients (62.9%), by evidence of malignant cells in cytology and/or DSOC-guided biopsies in 9 patients (25.7%), and by clinical and radiological follow-up in 4 patients (11.4%).

A benign etiology of IBS was found in 22 patients, confirmed by surgical pathology in one patient and by clinical and radiological follow-up in 21 patients. The mean follow-up was 18.2 ± 18.1 months; for patients with benign strictures, it was 24.1 ± 18.6 months, while for patients with malignant strictures, it was 14.5 ± 17.1 months.

### 3.1. Diagnostic Performance

Diagnostic performances of different techniques are shown in Table 2 and Figure 2.

DSOC showed a sensitivity of 85.7% (CI 95% 76.6–94.8%), a specificity of 95.5% (CI 95% 90–100%), and an overall diagnostic accuracy of 89.5% (CI 95% 81.5–97.4%); NPV was 80.8% (CI 95% 70.5–91%) and PPV was 96.8% (CI 95% 92.2–100%).

IDUS showed similar characteristics, with a sensitivity of 84.4% (CI 95% 74.5–94.2%), a specificity of 80% (CI 95% 69.1–90.9%), and an accuracy of 82.7% (CI 95% 72.4–93%). NPV and PPV were, respectively, 76.2% (64.6–87.8%) and 87.1% (CI 95% 78–96.2%).

DSOC-guided biopsies had lower sensitivity (63.6%, CI 95% 51.1–76.1%) and a specificity of 100% (83–100%); the diagnostic accuracy of targeted biopsies was 76.9% (CI 95% 66–87.9%).

Brush cytology showed a sensitivity of 51.6% (CI 95% 38.6–64.6%) and a specificity of 100% (CI 95% 68–100%). The accuracy of this technique was 61.5% (CI 95% 48.9–74.2%).

DSOC visualization and IDUS outperformed cytology in terms of sensitivity (*p* < 0.01, *p* = 0.03, *p* < 0.01, respectively) and accuracy (*p* < 0.01, *p* = 0.047, *p* = 0.03, respectively); NPV was significantly higher for DSOC and IDUS compared to cytology (*p* < 0.01). Finally, DSOC sensitivity was significantly higher than targeted biopsies (*p* = 0.05).

A comparison of the operating characteristics of each test is presented in Table 3.

### 3.2. Secondary Aim: Effect of Previous Stenting

Twenty-nine out of fifty-seven patients (50.9%) underwent the procedure with a previous stent in place, which was removed before starting DSOC; all but one were plastic stents. Twenty-eight patients (49.1%) did not have a stent in place.

The two groups were comparable in terms of age (66.5 ± 11.0 years vs. 68.0 ± 9.2, *p* = 0.58), gender (62.1% male and 75% male, respectively, *p* = 0.27), and etiology (malignant stricture in 55.2% vs. 67.9% of cases, respectively, *p* = 0.42).

The presence of a stent did not affect the diagnostic accuracy of DSOC visualization (89.7% in the stent group vs. 92.9% in the no-stent group, *p* > 0.99). However, the diagnostic accuracy of IDUS in stented patients showed a trend toward a slight decrease (74.1% vs. 92%), but the reduction was not statistically significant (*p* = 0.14).

### 3.3. Secondary Aim: Safety

Ten patients out of fifty-seven experienced a total of eleven AEs (17.5%); five were mild, five were moderate, and one was fatal. The most common AE was cholangitis (five cases, 8.8%), followed by acute pancreatitis (four cases, 7.0%) and one case of aspiration pneumonia. In one instance of cholangitis, a new endoscopic intervention was required to replace a malfunctioning plastic stent, while the others were managed with medical therapy. Hospital stay was extended by less than 7 days in all cases.

One patient with a history of coronary artery disease suffered a myocardial infarction four days after cholangioscopy, which was deemed unrelated to the endoscopic procedure. The patient underwent percutaneous transluminal coronary angioplasty and died during the hospitalization. All AEs are reported in Table 4.

## 4. Discussion

The goal of an accurate diagnosis of IBS is crucial due to the potentially vastly different prognosis based on etiology [14]. Conventional ERCP-based tissue acquisition methods have shown suboptimal diagnostic accuracy, prompting the emergence of DSOC as a promising innovation in IBS diagnosis [15]. Our results confirm these findings: DSOC showed nearly 90% diagnostic accuracy, significantly outperforming brush cytology (61.5%), which remains a standard practice in many endoscopy units.

Direct visualization during DSOC allows for the identification of endoscopic features typical of malignancy (such as papillary projections and tortuous vessels), achieving specificity comparable to histological specimens but with higher sensitivity.

Interestingly, optical diagnosis appears to offer better sensitivity and diagnostic accuracy compared to DSOC-guided biopsies. This “paradox” can be partially explained by the biological characteristics of cholangiocarcinoma, the predominant cause of IBS in our cohort: the desmoplastic nature, associated fibrosis, and submucosal spread of bile duct tumors may reduce the efficacy of superficial sampling methods. Additionally, cancers originating outside the bile duct, such as pancreatic cancers and metastatic tumors, are inherently more difficult to sample within the duct [16].

Moreover, the size and shape of the DSOC-dedicated forceps could contribute to the low sensitivity of targeted biopsy. As previously described [17,18], the small tissue sample collected by forceps must be carefully handled to avoid material loss during the standard formalin fixation and paraffin embedding. New dedicated processing protocols are needed to maximize the diagnostic yield [19].

Unlike EUS-guided tissue acquisition [20], DSOC lacks a standardized protocol of sampling, and the optimal number of biopsies to enhance diagnostic performance remains undefined. However, in a prospective study by Bang and colleagues, performing three biopsies achieved the correct diagnosis in 90% of cases in the absence of on-site cytopathology evaluation, comparable to the on-site approach [21]. The availability of “rapid on-site evaluation of touch imprint cytology” (ROSE-TIC) may further improve the diagnostic yield of DSOC-guided biopsies when accessible [22].

Our results align with existing literature, reporting DSOC visual accuracy ranging between 80 and 97% [7,23]. Initially, concerns were raised regarding a poor interobserver agreement for correctly classifying certain cholangioscopic features [24], often based more on investigators’ impressions than standardized, validated definitions [25,26]. The evolution of cholangioscopy to a digital platform with enhanced imaging quality [11], alongside the introduction of classification systems like the Monaco classification and the Robles–Medranda Criteria [27,28], has mitigated these concerns, as recently demonstrated by Kahaleh et al. [29].

Our study also demonstrated similar diagnostic efficacy for IDUS, with sensitivity and specificity exceeding 80%, significantly higher than cytology. IDUS also exhibited a trend towards higher sensitivity compared to targeted biopsies (84.4% vs. 63.6%, *p* = 0.09). Despite its potential benefits, the technique has not proliferated, with few endoscopists proficient in its application during ERCP. In our experience, IDUS provides a rapid and reliable assessment of strictures, residual stones, and ductal compression from extrinsic lesions, which are often challenging to assess by fluoroscopy.

Furthermore, it facilitates the precise assessment of the longitudinal extension of cholangiocarcinoma and provides an accurate evaluation of hepatic artery and portal vein infiltration, which is crucial for surgical planning [30,31]. Notably, IDUS is safer and more cost-effective than cholangioscopy, with reusable miniprobes capable of up to 50 examinations when properly handled, making it particularly beneficial in resource-limited settings [32].

An interesting finding from our study was the impact of biliary stenting on diagnostic accuracy. Diagnostic accuracy of DSOC remained comparable with or without a stent in place (89.7% vs. 92.9%), whereas IDUS diagnostic accuracy was slightly reduced in stented patients (74.1% vs. 92%). Although not statistically significant, this difference highlights how previous endoscopic manipulation of biliary strictures can alter biliary epithelium, potentially leading to inflammation and architectural distortion that can affect accuracy.

In addition to the diagnostic efficacy, we evaluated procedural safety. The AE rate in our study was 17.5%, falling in the wide range (1.7% to 25.4%) reported in previous studies [26,33,34]. Variability in AE rates across studies may stem from differing definitions of AEs, with higher rates in papers with more detailed definitions. Despite prophylactic antibiotic administration, cholangitis was the most common AE (8.8%), likely due to the intermittent irrigation necessary for adequate visualization of the biliary mucosa, facilitating retrograde bacterial flow in the biliary tree [35]. Currently, European guidelines consider cholangioscopy as a high-risk procedure for post-ERCP cholangitis, recommending antibiotic prophylaxis [36]. Further studies, including randomized controlled trials, are warranted to precisely define optimal measures for preventing this complication, potentially exploring a prolonged antibiotic course after cholangioscopy instead of single-dose prophylaxis.

Acute pancreatitis occurred in 7% of patients, higher than the 2–4% risk following standard ERCP [37].

Mechanical irritation and subsequent papillary swelling due to the cholangioscope passage, similar to the effects seen with a rigid catheter such as the IDUS miniprobe [38], may explain this increased risk. Interestingly, combining DSOC and IDUS in our cohort did not heighten the risk compared to previous DSOC studies [12,34]. Moreover, all observed cases of post-procedural acute pancreatitis were mild to moderate, managed conservatively, with minimal impact on hospital stay. Since post-procedural lipase levels were routinely determined in all patients, even mild cases of pancreatitis were less likely to be missed.

Several study limitations warrant consideration. Endoscopists were not blinded to previous imaging, laboratory results, and clinical history when evaluating biliary strictures, which potentially biased IDUS and DSOC visual assessment. Nonetheless, our patients were referred due to still indeterminate biliary strictures despite previous diagnostic attempts, reflecting real-world clinical practice. Additionally, this single-center study was conducted by experienced endoscopists in a high-volume referral hospital. While European guidelines recommend that IBS should be assessed and managed in tertiary referral centers [39,40], replicating our findings with less experienced endoscopists remains uncertain. Lastly, the relatively small sample size resulted in wide confidence intervals for our diagnostic operating characteristics, emphasizing the potential impact of one false negative or positive examination on the study outcome. Given the retrospective design of the study, a sample size calculation was not performed; however, based on our results, we estimated that including at least 74 patients would reveal significant differences in diagnostic accuracy between DSOC visual diagnosis and DSOC-targeted biopsy [41].

## 5. Conclusions

In conclusion, DSOC visualization and IDUS demonstrated optimal diagnostic yield in differentiating IBS. The high sensitivity of these techniques, compared to standard sampling methods, enabled the correct diagnosis of 90% of IBS in our cohort. A multimodal approach, with the possibility to perform different diagnostics in the same session with a tailored procedure, can assist endoscopists in the management of this challenging condition. DSOC showed the highest diagnostic accuracy and should be the method of choice in the evaluation of IBS. When DSOC is technically unfeasible due to the position or angulation of the stricture, IDUS can provide reliable results in over 80% of the patients, reducing the need for multiple procedures and shortening the time to reach an accurate diagnosis.

## Figures and Tables

**Figure 1 diagnostics-14-01316-f001:**
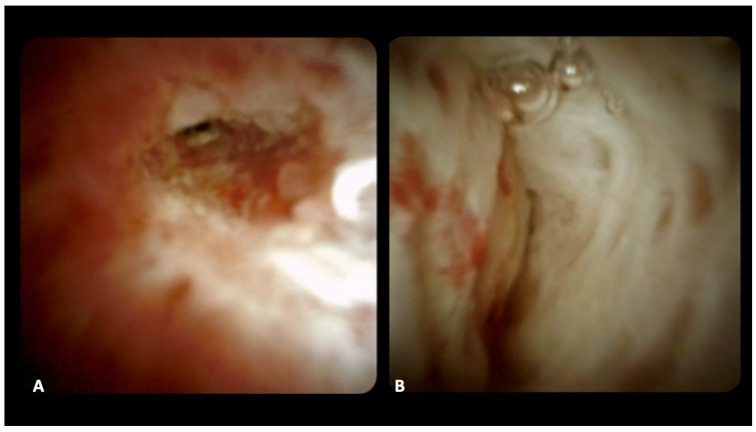
Cholangioscopic images of a malignant stricture with irregular mucosa and enlarged irregular vessels (**A**) and a benign stricture with smooth whitish biliary epithelium with fibrotic appearance (**B**).

**Figure 2 diagnostics-14-01316-f002:**
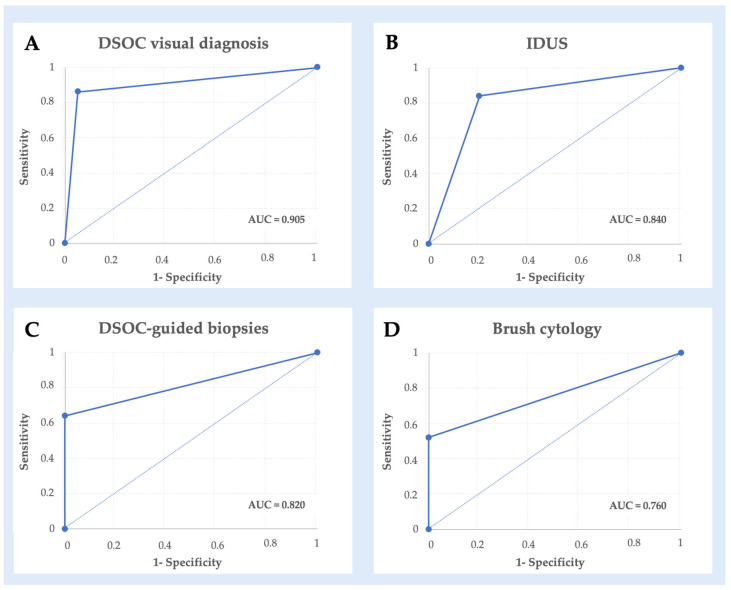
ROC curve of comparison between final diagnosis DSOC visual diagnosis (panel (**A**)), IDUS (panel (**B**)), DSOC-guided biopsies (panel (**C**)), and brush cytology (panel (**D**)).

**Table 1 diagnostics-14-01316-t001:** Baseline characteristics of patients.

Mean age (±SD), years	67.2 (±10.0)
Male/Female, *n* (%)/*n* (%)	39 (68.4%)/18 (31.6%)
Comorbidities, *n* (%)	34 (59.6%)
Cardiovascular	24 (42.1%)
Pulmonary	6 (10.5%)
Liver diseases	7 (12.3%)
Chronic kidney disease	1 (1.7%)
Type II diabetes mellitus	10 (17.5%)
Tobacco consumption	11 (19.3%)
Patients with previous ERCP, *n* %	39 (68.4%)
Mean previous ERCP (±SD)	1.97 ± 1.42
Patients with stent in place prior to DSOC, *n* (%)	29 (50.9%)
Previous biliary sphincterotomy, *n* (%)	38 (66.7%)
Previous cholecystectomy, *n* (%)	21 (36.8%)
**Stricture location, *n* (%)**	
Common bile duct	26 (45.6%)
Common hepatic duct	13 (22.8%)
Cystic duct	1 (1.7%)
Hepatic hilum	12 (21.1%)
Intrahepatic ducts	5 (8.8%)

SD, standard deviation; DSOC, direct single-operator cholangioscopy.

**Table 2 diagnostics-14-01316-t002:** Diagnostic yield of different techniques.

Techniques	Sensitivity (CI 95%)	Specificity (CI 95%)	Accuracy (CI 95%)	NPV (CI 95%)	PPV (CI 95%)
DSOC visualization	85.7% (76.6–94.8%)	95.5% (90.0–100%)	89.5% (81.5–97.4%)	80.8% (70.5–91%)	96.8% (92.2–100%)
IDUS	84.4% (74.5–94.2%)	80.0% (69.1–90.9%)	82.7% (72.4–93.0%)	76.2% (64.6–87.8%)	87.1% (78.0–96.2%)
DSOC targeted biopsy	63.6% (51.1–76.1%)	100% (83.0–100%)	76.9% (66.0–87.9%)	61.3% (48.6–73.9%)	100% (85.0–100%)
Brush cytology	51.6% (38.6–64.6%)	100% (78–100%)	61.5% (48.9–74.2%)	34.8% (22.4–47.1%)	100% (82–100%)

NPV, negative predictive value; PPV, positive predictive value; DSOC, digital single-operator cholangioscopy; IDUS, intraductal ultrasound.

**Table 3 diagnostics-14-01316-t003:** Comparison of diagnostic techniques.

	Sensitivity	Specificity	Accuracy	NPV	PPV
Comparison	*p* Value				
DSOC vs. IDUS	>0.99	0.17	0.41	0.73	0.35
DSOC vs. Biopsy	**0.05**	>0.99	0.12	0.15	>0.99
DSOC vs. Cytology	**<0.01**	>0.99	**<0.01**	**<0.01**	>0.99
IDUS vs. Biopsy	0.09	0.11	0.63	0.73	0.11
IDUS vs. Cytology	**<0.01**	0.29	**0.03**	**<0.01**	0.23
Biopsy vs. Cytology	0.45	>0.99	0.16	0.10	>0.99

NPV, negative predictive value; PPV, positive predictive value; DSOC, digital single-operator cholangioscopy; IDUS, intraductal ultrasound. Significant results (*p* < 0.05) are bold.

**Table 4 diagnostics-14-01316-t004:** Adverse events in the cohort.

Patient	Adverse Event	Severity Grade	Onset (Day after Procedure)	Management	Outcome
Patient 1	Cholangitis	Moderate	9	Endoscopic (repeated ERCP with stent replacement)	Favorable
Patient 2	Acute pancreatitis	Mild	1	Medical therapy	Favorable
Patient 2	Cholangitis	Moderate	2	Medical therapy	Favorable
Patient 3	Acute pancreatitis	Mild	0	Medical therapy	Favorable
Patient 4	Acute pancreatitis	Mild	0	Medical therapy	Favorable
Patient 5	Myocardial infarction	Fatal	4	Percutaneous transluminal coronary angioplasty	Fatal
Patient 6	Aspiration pneumonia	Moderate	0	Medical therapy	Favorable
Patient 7	Cholangitis	Moderate	1	Medical therapy	Favorable
Patient 8	Cholangitis	Mild	1	Medical therapy	Favorable
Patient 9	Cholangitis	Mild	1	Medical therapy	Favorable
Patient 10	Acute pancreatitis	Moderate	0	Medical therapy	Favorable

## Data Availability

The data that support the findings of this study are available from the corresponding author, M.S., upon reasonable request.

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
