# Peer review of "Direct Single-Operator Cholangioscopy and Intraductal Ultrasonography in Patients with Indeterminate Biliary Strictures: A Single Center Experience"

_diagnostics, 2024, doi:10.3390/diagnostics14131316_

Round 1

Reviewer 1 Report

Comments and Suggestions for Authors

This is a review of ID: diagnostics-3034947, “Direct Single Operator Cholangioscopy and Intraductal Ultrasonography in patients with Indeterminate Biliary Strictures: a single center experience.”. This manuscript is excellent in content, but needs some revisions.

 Major Comments:

Visual diagnosis by POCS is challenging due to the lack of a widely agreed-upon diagnostic definition. The definition of malignancy by endoscopic findings, considered most important in this manuscript, is what was defined in the previous generation of SPY Glass. The model used in this manuscript is the digital scope with improved resolution. On the other hand, the figure legend presented in the two examples provides detailed comments. As a countermeasure, the details of each observation in the cited literature should be tabulated separately for benign and malignant, or a definition of malignancy using another SPY Glass DS should be cited.

The direct visual biopsy with POCS is problematic because of its low diagnostic performance. The authors mention the possibility of ROSE-TIC as a solution to this problem. As was discussed when EUS-FNA was popularized, the specimen collection method (number of times, specimen processing method) should be considered first. For EUS-FNA, multiple punctures were recommended as one puncture was not sufficient. In addition to quickly immersing the collected specimens in formalin as a specimen processing method, various other methods were reported. Next is the pathological evaluation system.At a minimum, the number of biopsies performed on one stenosis in this study should be noted.

 Minor comments

1)     Page 2, line 59

"One shot" approach should be detailed.

2)     Page 6, line 169-170

In the three cases where specimens were not collected, what was being collected; as this is important information for specimen collection by POCS, it is best to provide as much detail as possible.

Author Response

Thank you for your valuable comments; a point-by-point response follows.

1) We have clarified the criteria for malignancy and benignity in the main text, and updated citations with paper using the digital version of cholangioscope (Spyglass DS).

2) We specified in the Methods section the minimum number of specimen collected and the fixation methods.

3) We tried to better explain in the introduction the concept of a "one shot" procedure, utilizing all different ancillary techniques during the same ERCP procedure to maximize the diagnostic efficacy.

4) We perform brush cytology as tissue acquisition method in the three patients with technical failure of DSOC-targeted biopsy; also IDUS and cholangioscopic visual evaluation was performed. We have now specified it more clearly in the text.

Reviewer 2 Report

Comments and Suggestions for Authors

The authors carried out a diagnostic study evaluating  DSOC and DSOC-targeted biopsies, intraductal ultrasound (IDUS) and brush cytology in patients with indeterminate 17 biliary stricture (IBS). The following queries are for authors considerations:

1. Please provide an estimation of sample size calculation that was carried out for this study. As the intention of authors is to compare four diagnostic procedures, n=57 tends to be very low. Additionally, provide a post-hoc power calculation with the estimates obtained from this study. This will also ensure that the study was adequately powered to detect the difference.

2. Please get the manuscript professionally edited by native English language speakers. For e.g. monocentric should be single-center, etc.

3. Consider providing ROC curves for each outcome comparing all the diagnostic procedures. It would visually (as well as statistically) appeal to the readers. 

Comments on the Quality of English Language

Moderate language edits are required.

Author Response

Thank you for your valuable comment, which allowed us to revise our paper, thereby enhancing its overall quality based on the provided feedback. Below is a point-by-point response to the queries raised.

1) Given the retrospective nature of the study, all consecutive cases within the study period were included and no prior sample size calculation was performed. We have highlighted this limitation of the study more clearly in the discussion, and provided a post-hoc sample size calculation to observe a significantly different accuracy between DSOC visualization and DSOC guided biopsies based on the study results, as suggested.

2)We conducted an extensive linguistic review of the paper to enhance its clarity and readability.

3)We have included figures of the ROC curves in the text, with the AUC evaluation of the different diagnostic methods

Round 2

Reviewer 2 Report

Comments and Suggestions for Authors

Thank you for the revision.